# Chip Placement with Diffusion Models

**Vint Lee** [1]  **Minh Nguyen** [1]  **Leena Elzeiny** [2]  **Chun Deng** [3]  **Pieter Abbeel** [1]  **John Wawrzynek** [1]

## Abstract

Macro placement is a vital step in digital circuit design that defines the physical location of large collections of components, known as macros, on a 2D chip. Because key performance metrics of the chip are determined by the placement, optimizing it is crucial. Existing learning-based methods typically fall short because of their reliance on reinforcement learning (RL), which is slow and struggles to generalize, requiring online training on each new circuit. Instead, we train a diffusion model capable of placing new circuits zero-shot, using guided sampling in lieu of RL to optimize placement quality. To enable such models to train at scale, we designed a capable yet efficient architecture for the denoising model, and propose a novel algorithm to generate large synthetic datasets for pre-training. To allow zero-shot transfer to real circuits, we empirically study the design decisions of our dataset generation algorithm, and identify several key factors enabling generalization. When trained on our synthetic data, our models generate high-quality placements on unseen, realistic circuits, achieving competitive performance on placement benchmarks compared to state-of-the-art methods.

## 1. Introduction

Placement is an important step of digital hardware design where collections of small components, such as logic gates (standard cells), and large design blocks, such as memories, (macros) are arranged on a 2-dimensional physical chip based on a connectivity graph (netlist) of the components. Because the physical layout of objects determines the length of wires (and where they can be routed), this step has a significant impact on key metrics, such as power consumption, performance, and area of the produced chip. In particular, the placement of macros, which is the focus of our work, is especially important because of their large size and high connectivity relative to standard cells.

Traditionally, macro placement is done with commercial tools such as Innovus from Cadence, which requires input from human experts. The process is also time-consuming and expensive. On the other hand, the use of ML techniques shows promise in automating this process, as well as creating better-optimized placements than commercial tools, which rely heavily on heuristics.

Even so, existing works mostly rely on reinforcement learning (RL) (Mirhoseini et al., 2020; Cheng & Yan, 2021; Lai et al., 2022; 2023; Gu et al., 2024), an approach with several key limitations. First, RL is challenging to scale — it is sample-inefficient, and has difficulty generalizing to new problems. Many of these methods, for instance, treat each new circuit as a separate task, training a new agent from scratch for every new netlist (Cheng & Yan, 2021; Lai et al., 2022; Gu et al., 2024). Despite efforts to mitigate this by incorporating offline pre-training, the scarcity of publicly-available data means that such methods struggle to generalize, and still require a significant amount of additional training for each new netlist (Mirhoseini et al., 2020; Lai et al., 2023). Second, by casting placement as a Markov Decision Process (MDP), these works require agents to learn a sequential placement of objects (standard cells or macros), which creates challenges when suboptimal choices near the start of the trajectory cannot be reversed.

To circumvent these issues, we instead adopt a different approach: leveraging powerful generative models, in particular diffusion models, to produce near-optimal chip placements for a given netlist. Diffusion models address the weaknesses of RL approaches because they can be trained offline at scale, then used zero-shot on new netlists, simultaneously placing all objects as shown in Figure 1. Moreover, our approach takes advantage of the great strides made in techniques for training and sampling diffusion models, such as guided sampling (Dhariwal & Nichol, 2021; Bansal et al., 2023), to achieve better results.

Training a large and generalizable diffusion model, however, comes with its own challenges. First, the vast majority of circuit designs and netlists of interest are proprietary,

---

[1]Department of EECS, UC Berkeley, CA, USA [2]SambaNova Systems Inc., Palo Alto, CA, USA [3]Computer Science Department, Stanford University, CA, USA. Correspondence to: Vint Lee <vint@berkeley.edu>.

*Proceedings of the $42^{nd}$ International Conference on Machine Learning*, Vancouver, Canada. PMLR 267, 2025. Copyright 2025 by the author(s).

severely limiting the quality and quantity of available training data. Second, many of these circuits are large, containing hundreds of thousands of macros and cells. The denoising model used must therefore be computationally efficient and scalable, in addition to working well within the noise-prediction framework.

Our work addresses these challenges, and we summarize our main contributions as follows:

**Synthetic Data Generation** We present a method for easily generating large amounts of synthetic netlist and placement data. Our insight is that the inverse problem — producing a plausible netlist such that a given placement is near-optimal — is much simpler to solve. This allows us to produce data without the need for commercial tools or higher-level design specifications.

**Dataset Design** We conduct an extensive empirical study investigating the generalization properties of models trained on synthetic data, identifying several factors, such as the scale parameter, that cause models to generalize poorly. We use these insights to design synthetic datasets that allow for effective zero-shot transfer to real circuits.

**Model Architecture** We propose a novel neural network architecture with interleaved graph convolutions and attention layers to obtain a model that is both computationally efficient and expressive.

By combining these ingredients, our method can generate placements for unseen netlists in a zero-shot manner, achieving results competitive with state-of-the-art on the IC-CAD04 (or IBM) (Adya et al., 2004) and ISPD2005 (Nam et al., 2005b) benchmarks. Remarkably, our model accomplishes this without ever having trained on real circuit data.

## 2. Related Work

Google's RL-based approach, CircuitTraining (Mirhoseini et al., 2021; Yue et al., 2022), employs a graph neural network (GNN) to generate netlist embeddings for multiple RL agents. While this method demonstrated state-of-the-art results, it requires computationally expensive online training on new circuits. Several RL approaches follow to improve on runtime (Cheng & Yan, 2021; Lai et al., 2022; Gu et al., 2024), macro ordering (Chen et al., 2023a), and proxy cost predictions (Zheng et al., 2023a;b; Wang et al., 2022; Ghose et al., 2021).

Several recent works have focused on reducing the long runtimes of RL methods. ChiPFormer (Lai et al., 2023) uses offline RL to reduce the amount of online training required. Despite strong results on several benchmarks, their method is constrained by the diversity and quantity of

data available, requiring hours of online training on each new netlist for good results, even on in-distribution circuits. EfficientPlace (Geng et al., 2024) reduces the exploration needed by combining global tree search with a local policy. Although this improved efficiency, their method still relies on online training for the local policy, leading to runtimes of several hours per placement.

In contrast, Flora (Liu et al., 2022a) and GraphPlanner (Liu et al., 2022b) deviate from sequential placement formulations by leveraging a variational autoencoder (VAE) (Kingma & Welling, 2022) to generate placements. Flora further introduces a synthetic data generation scheme; however, it lacks variation in object sizes and restricts connections to only the nearest neighbors, which, as our experiments indicate, limits generalization to realistic circuit layouts (see Section 5.1 and Table 4). Furthermore, their models struggle to learn the underlying distribution of legal placements, frequently producing overlapping results.

Other approaches avoid the use of machine learning altogether. WireMask-BBO (Shi et al., 2023) utilizes black-box optimization algorithms to find optimal macro placements over continuous coordinates, while legalizing and evaluating the solution quality on a discrete grid. However, their usage of black-box optimization, such as evolutionary algorithms, leads to lengthy search times that must be started from scratch for each new circuit.

## 3. Background

### 3.1. Problem Statement

Our goal is to train a diffusion model to sample from $f(x|c)$, where the placement $x$ is a set of 2D coordinates for each object and the netlist $c$ describes how the objects are connected in a graph, as well as the size of each object. We normalize the coordinates to the chip boundaries, so that they are within $[-1, 1]$.

We represent the netlist as a graph $(V, E)$ with node and edge attributes $\{p_i\}_{i \in V}$ and $\{q_{ij}\}_{(i,j) \in E}$. We define $p_i$ to be a 2D vector describing the normalized height and width of the object, while $q_{ij}$ is a 4D vector containing the positions of the source and destination pins, relative to the center of their parent object. We convert the netlist hypergraph into this representation by connecting the driving pin of each netlist to the others with undirected edges. This compact representation contains all the geometric information needed for placement, and allows us to leverage the rich body of existing GNN methods.

### 3.2. Evaluation Metrics

To evaluate generated placements, we use legality, which measures how easily the placement can be used for down-

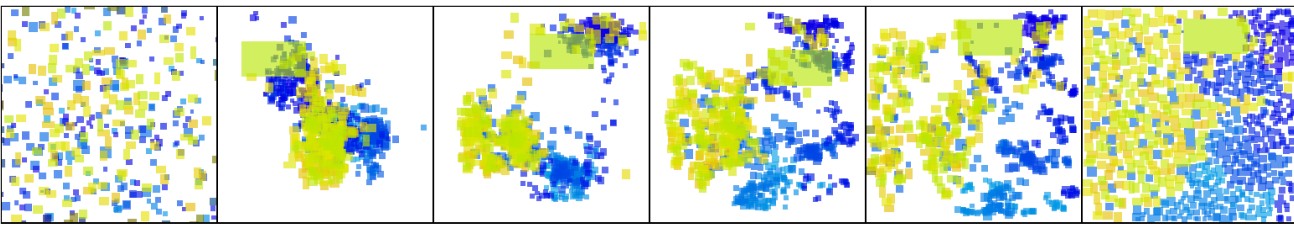

*Figure 1.* Denoising process for generating placements. In contrast to RL approaches, our method places all objects simultaneously. The middle 4 panels show the predicted output $\hat{x}_0$ at intervals of 200 steps, while the first and last panels are $x_T$ (Gaussian noise) and $x_0$ (generated placement).

stream tasks (eg. routing); and half-perimeter wire length (HPWL), which serves as a proxy for chip performance.

While a legal placement has to satisfy other criteria, in this work we focus on a simpler, commonly used constraint (Mirhoseini et al., 2020; Lai et al., 2023): the objects cannot overlap one another, and must be within the bounds of the canvas. We can therefore define *legality score* as $A_u/A_s$, where $A_u$ is the area, within the circuit boundary, of the union of all placed objects, and $A_s$ is the sum of areas of all individual objects. A legality of 1 indicates that all constraints are satisfied.

Routed wirelength influences critical metrics because long wires create delay between components, influencing timing and power consumption. HPWL is used as an approximation to evaluate placements prior to routing (Chen et al., 2006; Kahng & Reda, 2006). Because the scale of HPWL varies greatly between circuits, for our experiments on synthetic data we report the *HPWL ratio*, defined for a given netlist as $W_{\text{gen}}/W_{\text{data}}$, where $W_{\text{gen}}$ is the HPWL for the model-generated placement, while $W_{\text{data}}$ is the HPWL of the placement in the dataset.

Our objective is therefore to generate legal placements with minimal HPWL.

### 3.3. Diffusion Models

Diffusion models (Song et al., 2021; Ho et al., 2020) are a class of generative models whose outputs are produced by iteratively denoising samples using a process known as Langevin Dynamics. In this work we use the Denoising Diffusion Probabilistic Model (DDPM) formulation (Ho et al., 2020), where starting with Gaussian noise $x_T$, we perform $T$ denoising steps to obtain $x_{T-1}, x_{T-2}, \ldots, x_0$, with the fully denoised output $x_0$ as our generated sample. In DDPMs, each denoising step is performed according to

$$x_{t-1} = \alpha_t \cdot x_t + \beta_t \cdot \hat{\epsilon}_\theta(x_t, t, c) + \sigma_t \cdot z, \quad (1)$$

where $\alpha_t, \beta_t, \sigma_t$ are constants defined by the noise schedule, $z \sim \mathcal{N}(0, I)$ is injected noise, and $\hat{\epsilon}_\theta$ is the learned denoising model taking $x_t$, $t$ and context $c$ as inputs. By

training $\hat{\epsilon}_\theta$ to predict the noise added to samples from the dataset, DDPMs are able to model arbitrarily complex data distributions.

## 4. Methods

### 4.1. Generating Synthetic Data

We obtain datasets (Table 1) consisting of tuples $(x, c)$ using the method outlined below.

First, we randomly generate objects by sampling sizes uniformly and placing them at random within the circuit boundary, ensuring legality by retrying if objects overlap. Following Rent's Rule (Lanzerotti et al., 2005), we then sample a number of pins for each object using a power law.

To generate edges, we start by computing the distance $l$ for each pair of pins on different objects, then sample independently from Bernoulli($p(l)$), where $p(l) \in [0, 1]$ is a distance-dependent probability which we refer to as the edge distribution[1]. To approximate the structure of real circuits and bias the model towards lower HPWL placements, we choose $p \propto \exp(-l/s)$, so that the probability of generating edges decays exponentially with L1 distance $l$ normalized by a scale parameter $s$.

This simple algorithm, depicted in Figure 2, allows us to efficiently generate large numbers of training examples for our models without the using any commercial tools or design specifications. Using 32 CPUs, we are able to produce 100k "circuits" each containing approximately 200 objects in a day.

Our algorithm is also highly flexible, allowing many choices for the distributions of the number of objects, their sizes, the edges, scale parameters, and so on. To better understand the design space of synthetic datasets, we conduct an extensive empirical study in Section 5.1, identifying several factors that are vital for training models that transfer zero-shot to real circuits.

---

[1] $p$ is not, however, a probability distribution

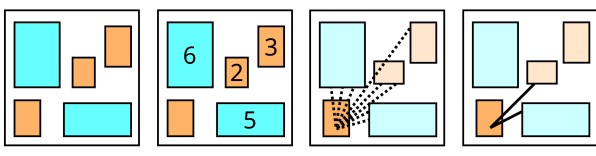

1. Place objects  2. Generate pins  3. Compute pin distances  4. Sample edges

*Figure 2.* Visualization of the steps involved in generating synthetic data.

Based on our results, we designed 2 synthetic datasets, *v1* and *v2*, with parameters listed in Table 1.

### 4.2. Model Architecture

We developed a novel architecture for the denoising model, shown in Figure 3. We highlight below several key elements of our design that we empirically determined (see Section 5.2) to be important for the placement task:

**Interleaved GNN and Attention Layers**  We use the message-passing GNN layers for their computational efficiency in capturing node neighborhood information, while the interleaved attention (Vaswani et al., 2017) layers address the oversmoothing problem in GNNs by allowing information transfer between nodes that are distant in the netlist graph, but close on the 2D canvas. We find that combining the two types of layers is critical, and significantly outperforms using either type alone.

**MLP Blocks**  We found that inserting residual 2-layer MLP blocks between each GNN and Attention block improved performance significantly for a negligible increase in computation time.

**Sinusoidal 2D Encodings**  The model receives 2D sinusoidal position encodings, in addition to the original $(x, y)$ coordinates, as input. This method improves the precision with which the model places small objects, leading to placements with better legality.

In this work, we use 3 sizes of models: *Small*, *Medium*, and *Large*, with 233k, 1.23M, and 6.29M parameters respectively. A full list of model hyperparameters can be found in Appendix A.

### 4.3. Guided Sampling

One key advantage of using diffusion models is the ease of optimizing for downstream objectives through guided sampling. We use backwards universal guidance (Bansal et al., 2023) with easily computed potential functions to optimize the generated HPWL and legality without training additional reward models or classifiers. The guidance potential $\varphi(x)$ is defined as the weighted sum $w_{\text{legality}} \cdot \varphi_{\text{legality}} + w_{\text{hpwl}} \cdot \varphi_{\text{hpwl}}$

of potentials for each of our optimization objectives.

The legality potential $\varphi_{\text{legality}}(x)$ for a netlist with objects $V$ is given by:

$$\varphi_{\text{legality}}(x) = \sum_{i,j \in V} \min(0, d_{ij}(x))^2 \qquad (2)$$

where $d_{ij}$ is the signed distance between objects $i$ and $j$, which we can compute easily for rectangular objects. Note that the summand is 0 for any pair of non-overlapping objects, and increases as overlap increases.

We define $\varphi_{\text{hpwl}}(x)$ simply as the HPWL of the placement $x$. We compute this in a parallelized, differentiable manner by casting HPWL computation in terms of the message-passing framework used in GNNs (Gilmer et al., 2017) and implementing a custom GNN layer with no learnable parameters in PyG (Fey & Lenssen, 2019).

Instead of gradients from a classifier (Dhariwal & Nichol, 2021), we use the backwards universal guidance force $g(x_t) = \Delta_\varphi \hat{x}_0$. Here, $\hat{x}_0$ is the prediction of $x_0$ based on the denoising model's output at time step $t$, and $\Delta_\varphi$ is the $\varphi$-optimal change in $\hat{x}_0$-space, computed using gradient descent. The combined diffusion score is then given by $f_\theta(x_t) + w_g \cdot g(x_t)$. We refer the reader to Bansal et al. (2023) for more details.

In the simple implementation, $w_{\text{legality}}$ and $w_{\text{hpwl}}$ are set as constant hyperparameters. However, the optimal weights can vary depending on the circuit's connectivity properties. Instead, we take inspiration from constrained optimization to automatically tune the weights. To solve

$$\min_x \varphi_{\text{hpwl}}(x) \qquad \text{s.t.} \qquad \varphi_{\text{legality}}(x) = 0, \qquad (3)$$

we optimize the Lagrangian $\mathcal{L}(\lambda, x) = \varphi_{\text{hpwl}}(x) + \lambda \cdot \varphi_{\text{legality}}(x)$ simultaneously with respect to $x$ and the Lagrange multiplier $\lambda$. We instantiate this idea during guidance by performing interleaved gradient descent steps of $\varphi_{\text{hpwl}}(x) + w_{\text{legality}} \cdot \varphi_{\text{legality}}(x)$ with respect to $x$, and $w_{\text{legality}} \cdot (\varphi_{\text{legality}}(x) - \varepsilon)$ with respect to $w_{\text{legality}}$.

### 4.4. Training and Evaluation

Due to the lack of real placement data, we train our models entirely on synthetic data (see Section 4.1). For placing real circuit netlists, we train our models in two stages: we first train on our *v1* dataset of smaller circuits, then fine-tune on *v2* which contains larger circuits. Details on dataset design are provided in Section 5.1.

We evaluate the performance of our model on circuits in the publicly available ICCAD04 (Adya et al., 2004) (also known as IBM) and ISPD2005 (Nam et al., 2005a) benchmarks. Because these circuits contain hundreds of thousands of

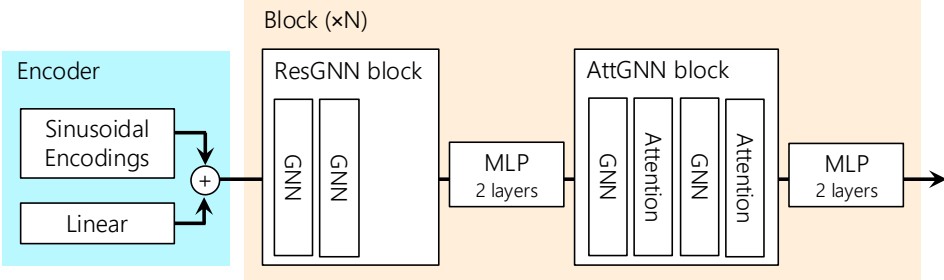

*Figure 3.* Diagram of our denoising model. Residual connections, edge feature inputs, nonlinearities, and normalization layers have been ommitted for clarity.

small standard cells, we follow prior work (Mirhoseini et al., 2020) and cluster the standard cells into 512 partitions using hMetis (Karypis et al., 1997). Each cluster is assigned to the nets of its constituent standard cells (nets within a single cluster are removed) with pins located at the cluster center, while the size of each cluster is the total area of its standard cells.

### 4.5. Implementation

Our models are implemented using Pytorch (Paszke et al., 2019) and Pytorch-Geometric (Fey & Lenssen, 2019), and trained on machines with Intel Xeon Gold 6326 CPUs, using a single Nvidia A5000 GPU. We train our models using the Adam optimizer (Kingma & Ba, 2014) for 3M steps, with 250k steps of fine-tuning where applicable.

## 5. Experiments

### 5.1. Designing Synthetic Data

To generate synthetic datasets that allow for zero-shot transfer, we first have to understand which parameters are important for generalization, and which ones are not. We therefore investigate the generalization capabilities of our model along several axes by evaluating a single trained model on datasets generated using various parameters. By identifying parameters that the model struggles to generalize across, we can design our synthetic dataset to facilitate zero-shot transfer by ensuring that for such parameters, the synthetic distribution covers that of real circuits (see Section 5.1.4).

In this section, we evaluate a model trained on a dataset with a narrow distribution, with key parameters listed in Table 1. The full set of parameters is listed in Appendix A.

#### 5.1.1. NUMBER OF EDGES AND VERTICES

Figure 4 shows how legality changes with the number of edges in the test dataset. The model generalizes remarkably well to datasets with more edges than it was trained on, while performance degrades quickly when fewer edges are present. We hypothesize that an increased number of edges allows

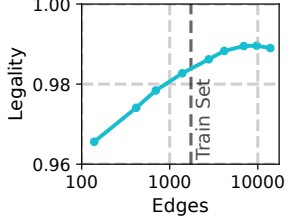 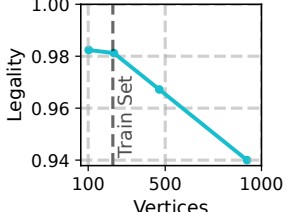

*Figure 4.* Legality decreases on circuits with fewer edges, while adding edges does not degrade performance. The training data has 1740 edges on average.

*Figure 5.* Legality decreases on circuits with more vertices, indicating poor generalization. The training data has 230 edges on average.

the GNN layers to propagate information more efficiently, improving or maintaining the model's performance.

In contrast, Figure 5 shows that the model struggles to generalize to larger circuits than it was trained on, with legality decreasing as the number of vertices increases.

#### 5.1.2. SCALE PARAMETER

In our data generation algorithm, the scale parameter $s$ determines the expected length of generated edges. A larger $s$ means that distant pins are more likely to be connected, while a small value of $s$ means that only nearby pins are connected. Thus, the scale parameter has a significant impact on the properties of the graph generated, such as the number of neighbors per vertex, as well as the optimality of the corresponding placement. To understand how these effects influence the placements generated by the model, we evaluate our model, trained on data with a fixed value of $s$, on datasets with different scale parameters.

Figure 6 shows that the model performs well at longer scale parameters up to $s = 0.4$, with legality dropping sharply past it. Meanwhile, lowering $s$ causes HPWL to worsen significantly, with the model generating placements more than $1.5\times$ worse than the dataset.

This could be because for small $s$, the presence of edges

*Table 1.* Parameters of datasets used. For *v1* and *v2*, we sample $s$ for each circuit from a log-uniform distribution, then compute a scale-dependent multiplier $\gamma(s)$ to control the number of edges in each circuit.

| Name | Circuits | Vertices | Edges | Scale Parameter ($s$) | $p(l)$ |
|------|----------|----------|-------|----------------------|--------|
| *v0* | 40000 | 230 | 1740 | 0.2 | $\gamma \cdot \exp\left(-l/s\right)$ |
| *v1* | 40000 | 230 | 1600 | $\sim \log U(0.05, 1.6)$ | $\gamma(s) \cdot \exp\left(-l/s\right)$ |
| *v2* | 5000 | 960 | 9510 | $\sim \log U(0.025, 0.8)$ | $\gamma(s) \cdot \exp\left(-l/s\right)$ |

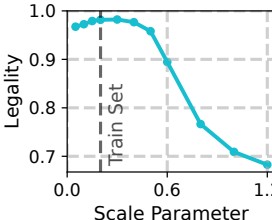 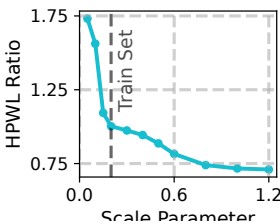

*Figure 6.* Legality drops sharply when increasing scale parameter past a certain point, while scale parameters smaller than the training data causes HPWL to worsen significantly. The training data is generated using a scale of 0.2.

*Table 2.* Test performance is near-identical to training performance on datasets using various $p(l)$. $\sigma$ is the sigmoid function, and $s$ is the scale parameter, which we choose such that the mean edge length matches that of training data.

| $p(l) \propto$ | Exponential $e^{(-l/s)}$ | Sigmoid $\sigma(l - s)$ | Linear $\max(\frac{s-x}{s}, 0)$ |
|----------------|--------------------------|-------------------------|----------------------------------|
| Legality | 0.982 | 0.983 | 0.982 |
| HPWL Ratio | 1.003 | 1.012 | 1.003 |

In Table 2, we see that our model performs well, both in legality and HPWL, on circuits where edges are sampled from different distributions. This is a promising indication that our training data can allow models to generalize zero-shot to unseen circuits that lie outside the training distribution.

### 5.1.4. ZERO-SHOT TRANSFER TO REAL CIRCUITS

These results allow us to design new datasets, which we refer to as *v1* and *v2*, that better enable zero-shot transfer to real circuits. Because our models generalize poorly to larger circuits, they require training on circuits similar in size to real (clustered) circuits, which contain $\sim$1000 components. To satisfy this while maintaining computational efficiency, we pre-train on a large set of smaller circuits (*v1*) before fine-tuning on a small set of larger circuits (*v2*), keeping the number of edges low in both datasets. To ensure model performance across different length scales, we also train on a broad distribution of scale parameters. Finally, since an exponentially decaying $p$ generalizes well to other distributions without sacrificing HPWL, we continue using it in our new datasets.

The parameters for generating the *v1* and *v2* datasets are summarized in Table 1, with a full list provided in Table 8.

We find in Table 4 that training on these broad-distribution datasets allow for effective zero-shot transfer to the real-world circuits in the IBM benchmark, with legality increasing significantly when training on *v1* and *v2*, compared to the narrower *v0* dataset.

between objects means that they are very likely near each other in the dataset placement, providing the model with a lot of information on where objects should be placed. If the model is then evaluated on circuits with larger $s$, the increased number of neighbors causes the model to place too many objects in the same vicinity, leading to clumps forming (see Figure 7) and poor legality. Conversely, when evaluated on circuits with smaller $s$, the model's inductive biases are not strong enough to place connected objects as close together as possible, leading to longer, worse, HPWL.

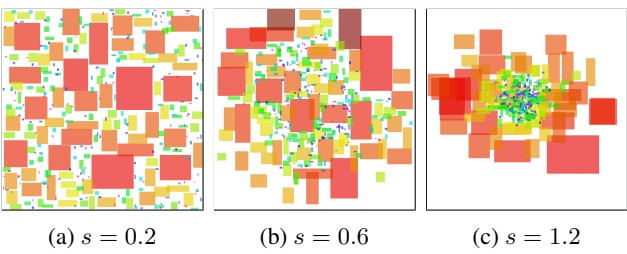

(a) $s = 0.2$     (b) $s = 0.6$     (c) $s = 1.2$

*Figure 7.* Increasing scale parameter causes the diffusion model, trained only on circuits with $s = 0.2$, to clump objects together.

### 5.1.3. DISTRIBUTION OF EDGES

To bias the model towards generating placements with low HPWL, we sample edges with probability $p(l)$ that exponentially decays with edge length $l$. However, there is no guarantee that optimal placements for real circuits follow such a distribution. We therefore investigate how this choice of $l$ affects generalization to other edge distributions $p(l)$.

Moreover, models trained on our synthetic dataset far outperform those trained on the existing data generation algorithm proposed by Flora (Liu et al., 2022a;b). We see in Table 4

that our *v1* dataset achieves a legality of 0.821, far higher than Flora's 0.264, demonstrating that our dataset allows for much better zero-shot generalization.

Thus, although our simple data generation algorithm does not capture all the nuances of real circuits, such as the multimodal distributions of both edges and object sizes, it covers the important features well enough for our model to learn to produce reasonable placements on IBM circuits, some of which are shown in Figure 8.

## 5.2. Model Architecture

We demonstrate the importance of several components of our model architecture through ablations, shown in Table 3. When either the sinusoidal encodings or MLP blocks are removed, the model performs substantially worse in both legality and HPWL. Replacing attention layers with graph convolutions also causes sample quality to plummet, as evidenced by poor legality scores.

*Table 3.* Legality and HPWL of various models on the *v1* dataset after 1M steps. Our models scale favorably, and our ablations validate the importance of several components of our model.

| Model | #Param. | Legality | HPWL Ratio |
|---|---|---|---|
| *Small* | 0.233M | 0.948 | 1.072 |
| *Medium* | 1.23M | 0.960 | 1.039 |
| – No attention | 1.42M | 0.799 | 1.035 |
| – No MLP | 0.698M | 0.946 | 1.060 |
| – No encodings | 1.21M | 0.949 | 1.061 |
| *Large* | 6.29M | **0.976** | **1.032** |

Our model also exhibits favorable scaling properties, with Table 3 showing significant and monotonic improvements in model performance (both legality and HPWL) with increasing model size. This suggests scaling up models as an attractive strategy for improving performance on more complex datasets, particularly since synthetic data is unlimited.

## 5.3. Guided Sampling on Real Circuits

To determine the effectiveness of guidance in improving sample quality, we used our model to generate placements for the IBM benchmark with standard cells clustered.

As shown in Table 4, guidance dramatically improves legality and HPWL during zero-shot sampling, with legality increasing to nearly 1 while simultaneously shortening HPWL by 7.1%. This result shows that our guidance method is effective in optimizing generated samples without requiring additional training. An example of the generated placements with and without guidance is shown in Figure 8.

Moreover, we find that our placements are significantly

*Table 4.* Average HPWL and legality achieved by our models on the clustered IBM benchmark. DREAMPlace figures, and results for a model trained on Flora's dataset are included for comparison.

| Model | Legality | HPWL ($10^7$) |
|---|---|---|
| DREAMPlace | - | 3.724 |
| *Large+Flora* | 0.2640 | 3.740 |
| *Large+v0* | 0.7794 | 3.252 |
| *Large+v1* | 0.8213 | 3.281 |
| *Large+v2* | 0.8835 | 3.203 |
| *Large+v2* (Guided) | **0.9970** | **2.976** |

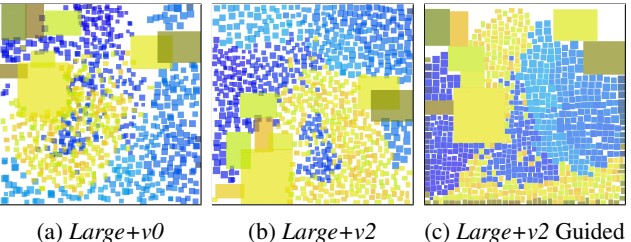

(a) *Large+v0*     (b) *Large+v2*     (c) *Large+v2* Guided

*Figure 8.* While our $Large$ model, when trained on $v2$ data, can produce reasonable placements, guided sampling improves quality significantly. ibm04 is shown here, with macros in yellow, while standard cell clusters are colored blue.

better than those produced by the state-of-the-art DREAM-Place (Lin et al., 2019; Chen et al., 2023b), with 20% lower HPWL on average. While we note that DREAMPlace, a mixed-size placer, is not optimized for placing clustered circuits, this result is nevertheless a strong indication of our method's ability to generate high-quality placements for real circuits. This is especially remarkable, showing that **models trained entirely on synthetic data can transfer effectively to real circuits in a zero-shot manner**.

## 5.4. Macro-only Placement of Real Circuits

To compare our method to existing macro placement techniques, we also generated placements in the macro-only setting by removing all standard cells from the IBM and ISPD circuit netlists. For brevity, we present only averages for each benchmark in this section, with results for individual circuits found in Appendix B.

We see from Table 5 that our method outperforms existing macro placers by a large margin, reducing average HPWL by over 50% compared to prior works in both the IBM and ISPD benchmarks.

Moreover, Table 5 also shows that placements generated by our method have far lower congestion that those produced by existing methods. Here, we measure the RUDY congestion (Spindler & Johannes, 2007) using the method in Shi et al. (2023). The strong performance of our method on the

congestion metric, despite not explicitly optimizing for it, is consistent with Shi et al. (2023)'s observation that RUDY congestion and HPWL are positively correlated.

*Table 5.* Average HPWL and RUDY congestion of our model compared to various baselines on the IBM and ISPD benchmarks in the macro-only setting. Our model significantly outperforms existing methods, achieving much lower HPWL and congestion.

| Method | HPWL ($10^5$) | Congestion |
|---|---|---|
| **IBM Average** | | |
| MaskPlace | 8.72 | 345 |
| WireMask-BBO | 7.43 | 324 |
| ChiPFormer | 7.33 | 336 |
| EfficientPlace | 8.32 | 367 |
| Diffusion (Ours) | **2.49** | **196** |
| **ISPD Average** | | |
| MaskPlace | Timeout on 2/8 circuits. | |
| WireMask-BBO | 154 | 1485 |
| ChiPFormer | 116 | 836 |
| Diffusion (Ours) | **45.9** | **546** |

### 5.5. Mixed-Size Placement of Real Circuits

While our results (Table 4 and Table 5) have shown that our method can produce high-quality macro placements for clustered and macro-only circuits, we also wish to investigate if these macro placements are useful for downstream tasks, particularly for mixed-size placement.

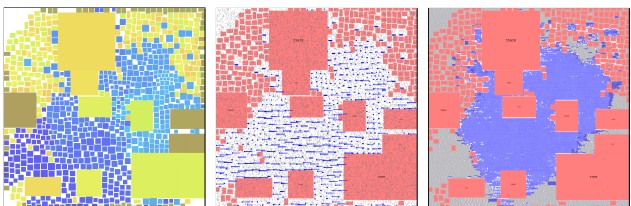

(a) Diffusion model placement.

(b) Initialize and fix macros.

(c) Place standard cells.

*Figure 9.* Our model can be used for mixed-size placement by first placing macros and clusters, then using our placements as inputs to a standard cell placer. ibm03 is shown here.

To perform mixed-size placement (which requires placing all standard cells and macros) using our method, we first use our diffusion models to place clusters and macros (Figure 9a). We then initialize the position of each standard cell to the position of the corresponding cluster, and copy the positions of the macros (Figure 9b). Finally, we use DREAMPlace 4.1 (Chen et al., 2023b) to place the standard cells, thus obtaining a full placement of standard cells and macros (Figure 9c). To elucidate the impact of our macro placements on the final placement quality, we keep

the macro positions fixed during standard cell placement. This is in line with earlier works (Lai et al., 2022; Shi et al., 2023; Mirhoseini et al., 2020; Cheng & Yan, 2021) but contrasts with Lai et al. (2023), where the contribution of the macro placement technique is not as clear because the macros move significantly during standard cell placement.

In Table 6, we compare results from our method to other macro placement baselines, as well as DREAMPlace. The baselines include both the learning-based ChiPFormer (Lai et al., 2023) and MaskPlace (Lai et al., 2022), as well as the learning-free WireMask-BBO[2] (Shi et al., 2023). All of these baselines use DREAMPlace for standard cell placement. We find that our method outperforms prior macro placement methods by a wide margin, while improving over DREAMPlace by $4\%$ on average. This indicates that the strong performance of our model on clustered macro placement transfers well to mixed-size placements. Moreover, our method, by design, can be easily applied zero-shot to new circuits and takes minutes to run (see Table 13), while other methods require RL fine-tuning or black box optimization, spending hours on each new circuit.

## 6. Conclusion

In this work, we explored an approach that departs from many existing methods for tackling macro placement: using diffusion models to generate placements. To train and apply such models at scale, we developed a novel data generation algorithm, designed synthetic datasets that enable zero-shot transfer to real circuits, and designed a neural network architecture that performs and scales well. We show that when trained on our synthetic data, our models generalize to new circuits, and when combined with guided sampling, can generate optimized placements even on large, real-world circuit benchmarks.

Even so, our work is not without limitations. RL methods, while slow, provide a means of trading test-time compute for better sample quality. We believe applying such methods, through DDPO (Black et al., 2024) for instance, could combine the strengths of generative modeling and RL fine-tuning. We also note that our synthetic data does not capture all the nuances of real data, such as multimodal edge distributions, and believe this is an interesting area for further study.

In conclusion, we find that training diffusion models on synthetic data is a promising approach, with our models generating competitive placements despite never having trained on realistic circuit data. We hope that our results inspire further work in this area.

---

[2]WireMask-BBO fails to place ibm12 and ibm15 so a lower-bound average is computed by substituting the smallest HPWL in the corresponding rows

*Table 6.* Comparison of HPWL ($10^6$) averaged over 5 seeds, using various techniques for mixed-size placement, on the IBM benchmark.

| Circuit | MaskPlace + DP | WireMask-BBO + DP | ChiPFormer + DP | DREAMPlace | Diffusion (Ours) |
|---|---|---|---|---|---|
| ibm01 | 3.33 | 2.84 | 3.35 | 2.23 | 2.09 |
| ibm02 | 7.30 | 6.87 | 6.24 | 5.79 | 4.43 |
| ibm03 | 10.1 | 9.81 | 10.9 | 10.4 | 7.30 |
| ibm04 | 10.4 | 9.65 | 10.1 | 9.13 | 8.00 |
| ibm05 | 7.67 | 7.67 | 7.67 | 7.60 | 7.79 |
| ibm06 | 7.62 | 8.41 | 7.76 | 6.15 | 8.31 |
| ibm07 | 13.3 | 13.0 | 13.4 | 11.1 | 9.60 |
| ibm08 | 15.5 | 15.9 | 15.7 | 12.3 | 13.3 |
| ibm09 | 16.2 | 15.4 | 16.9 | 12.8 | 12.6 |
| ibm10 | 46.8 | 45.2 | 45.4 | 44.8 | 30.2 |
| ibm11 | 23.5 | 24.6 | 23.6 | 16.6 | 17.3 |
| ibm12 | 46.1 | Failed | 48.8 | 31.0 | 34.0 |
| ibm13 | 28.2 | 28.0 | 28.4 | 23.2 | 23.0 |
| ibm14 | 45.4 | 48.2 | 46.5 | 31.3 | 34.5 |
| ibm15 | 53.4 | Failed | 55.8 | 51.3 | 45.0 |
| ibm16 | 65.9 | 63.2 | 67.3 | 53.0 | 52.7 |
| ibm17 | 72.9 | 69.7 | 71.4 | 57.9 | 60.4 |
| ibm18 | 42.2 | 41.6 | 41.1 | 37.6 | 38.6 |
| Average | 28.7 | 27.0 | 28.9 | 23.6 | 22.7 |

## Acknowledgements

This work was supported in part by an ONR DURIP grant and the BAIR Industrial Consortium. Pieter Abbeel holds concurrent appointments as a Professor at UC Berkeley and as an Amazon Scholar. This paper describes work performed at UC Berkeley and is not associated with Amazon.

## Impact Statement

This paper presents work whose goal is to advance the field of Machine Learning. There are many potential societal consequences of our work, none which we feel must be specifically highlighted here.

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

# A. Implementation Details

Hyperparameters for our model are listed in Table 7. We used the DDPM formulation found in Ho et al. (2020) with a cosine noise schedule over 1000 steps. We also list parameters for generating our synthetic datasets in Table 8.

*Table 7.* Hyperparameters for our models. We refer the reader to our code for more details.

|  | *Small* | *Medium* | *Large* |
|---|---|---|---|
| **Model Dimensions** | | | |
| Model size | 64 | 128 | 256 |
| Blocks | 2 | 2 | 3 |
| Layers per block | 2 | | |
| AttGNN size | 32 | 32 | 256 |
| ResGNN size | 64 | 256 | 256 |
| MLP size factor | 4 | | |
| MLP layers per block | 2 | | |
| | | | |
| **Input Encodings** | | | |
| Timestep encoding dimension | 32 | | |
| Input encoding dimension | 32 | | |
| | | | |
| **GNN Layers** | | | |
| Type | GATv2 (Brody et al., 2022) | | |
| Heads | 4 | | |
| | | | |
| **Guidance Parameters** | | | |
| $w_{\text{HPWL}}$ | 0.0001 | | |
| $\hat{x}_0$ optimizer | SGD | | |
| $\hat{x}_0$ optimizer learning rate | 0.008 | | |
| $w_{\text{legality}}$ optimizer | Adam | | |
| $w_{\text{legality}}$ optimizer learning rate | 0.0005 | | |
| $w_{\text{legality}}$ initial value | 0 | | |
| Gradient descent steps | 10 | | |
| $\varepsilon$ | 0.0001 | | |

*Table 8.* Parameters for generating our synthetic data. Unless otherwise stated, *v1* and *v2* use the same parameters as *v0*. We refer the reader to our code for more details. Sizes and distances are normalized so that a value of 2 corresponds to the width of the canvas.

| | *v0* | *v1* | *v2* |
|---|---|---|---|
| **Stop Density Distribution** | | | |
| Distribution type | Uniform | | |
| Low | 0.75 | | |
| High | 0.9 | | |
| | | | |
| **Aspect Ratio Distribution** | | | |
| Distribution type | Uniform | | |
| Low | 0.25 | | |
| High | 1.0 | | |
| | | | |
| **Object Size Distribution** | | | |
| Distribution type | Clipped Exponential | | |
| Scale | 0.08 | | 0.04 |
| Max | 1.0 | | 0.5 |
| Min | 0.02 | | 0.01 |
| | | | |
| **Edge Distribution** | | | |
| Distribution $p(l)$ | $\gamma \cdot \exp\left(-l/s\right)$ | | |
| Scale $s$ | 0.2 | $\sim \log U(0.05, 1.6)$ | $\sim \log U(0.025, 0.8)$ |
| Max $p$ | 0.9 | | |
| $\gamma$ | 0.21 | $0.212 \cdot s^{-1.42}$ | $0.00792 \cdot s^{-1.42}$ |

## B. Additional Results

We present results for macro placements for each circuit in the IBM and ISPD benchmarks below. HPWL figures are shown in Table 9 and Table 10, congestion in Table 11 and Table 12, and runtime in Table 13 and Table 14.

In some cases, our values differ from those in the original papers because some baselines select and place only a subset of the macros, with differing selection criteria among baselines, while we place all macros for all baselines to ensure a fair comparison.

*Table 9.* Comparison of HPWL ($10^5$) for macro-only placements on the IBM benchmark. ibm05 has been omitted because it contains no macros.

| Circuit | MaskPlace | WireMask-BBO | ChiPFormer | EfficientPlace | Diffusion (Ours) |
|---------|-----------|--------------|------------|----------------|------------------|
| ibm01 | 4.30 | 2.78 | 3.88 | 3.66 | 1.16 |
| ibm02 | 5.54 | 4.19 | 5.05 | 4.42 | 2.68 |
| ibm03 | 3.31 | 3.30 | 3.74 | 3.87 | 1.07 |
| ibm04 | 6.91 | 5.43 | 5.96 | 6.10 | 2.40 |
| ibm06 | 0.93 | 0.85 | 0.87 | 0.84 | 0.32 |
| ibm07 | 2.67 | 2.66 | 2.36 | 3.42 | 0.78 |
| ibm08 | 20.6 | 19.2 | 19.9 | 19.3 | 9.32 |
| ibm09 | 2.45 | 1.76 | 1.77 | 2.57 | 0.44 |
| ibm10 | 23.8 | 18.2 | 18.2 | 20.6 | 5.28 |
| ibm11 | 4.15 | 3.75 | 3.25 | 4.70 | 0.78 |
| ibm12 | 14.9 | 11.8 | 13.0 | 12.1 | 2.85 |
| ibm13 | 4.58 | 4.41 | 4.02 | 5.37 | 1.05 |
| ibm14 | 8.43 | 9.80 | 7.44 | 11.7 | 2.42 |
| ibm15 | 4.68 | 7.77 | 2.67 | 5.98 | 1.06 |
| ibm16 | 18.3 | 14.8 | 15.5 | 15.2 | 6.11 |
| ibm17 | 16.8 | 12.2 | 13.7 | 17.9 | 3.20 |
| ibm18 | 5.98 | 3.44 | 4.19 | 3.64 | 1.52 |
| Average | 8.72 | 7.43 | 7.33 | 8.32 | 2.49 |

*Table 10.* Comparison of HPWL ($10^5$) for macro-only placements on the ISPD benchmark.

| Circuit | MaskPlace | WireMask-BBO | ChiPFormer | Diffusion (Ours) |
|---------|-----------|--------------|------------|------------------|
| adaptec1 | 8.57 | 5.81 | 6.75 | 9.19 |
| adaptec2 | 77.7 | 54.5 | 63.8 | 31.0 |
| adaptec3 | 108 | 59.2 | 73.2 | 54.4 |
| adaptec4 | 91.9 | 62.7 | 85.8 | 54.5 |
| bigblue1 | 3.11 | 2.12 | 3.05 | 2.64 |
| bigblue2 | Timeout | 186 | 85.8 | 38.8 |
| bigblue3 | 84.0 | 66.2 | 79.2 | 35.9 |
| bigblue4 | Timeout | 798 | 548 | 141 |
| Average | — | 154 | 116 | 45.9 |

*Table 11.* Comparison of congestion (RUDY estimator) for macro-only placements on the IBM benchmark. ibm05 has been omitted because it contains no macros.

| Circuit | MaskPlace | WireMask-BBO | ChiPFormer | EfficientPlace | Diffusion (Ours) |
|---------|-----------|--------------|------------|----------------|------------------|
| ibm01 | 289 | 253 | 266 | 316 | 160 |
| ibm02 | 228 | 243 | 205 | 257 | 178 |
| ibm03 | 176 | 173 | 173 | 214 | 117 |
| ibm04 | 449 | 483 | 490 | 480 | 260 |
| ibm06 | 79.2 | 77.1 | 76.9 | 76.7 | 42.8 |
| ibm07 | 154 | 164 | 160 | 177 | 83.0 |
| ibm08 | 1232 | 1198 | 1261 | 1288 | 776 |
| ibm09 | 127 | 119 | 111 | 153 | 49.1 |
| ibm10 | 480 | 463 | 466 | 538 | 362 |
| ibm11 | 180 | 183 | 172 | 240 | 69.2 |
| ibm12 | 392 | 212 | 357 | 360 | 190 |
| ibm13 | 163 | 202 | 177 | 209 | 86.0 |
| ibm14 | 378 | 375 | 378 | 418 | 232 |
| ibm15 | 162 | 173 | 173 | 227 | 69.2 |
| ibm16 | 574 | 497 | 528 | 534 | 334 |
| ibm17 | 531 | 464 | 488 | 483 | 204 |
| ibm18 | 271 | 221 | 229 | 266 | 111 |
| Average | 345 | 324 | 336 | 367 | 196 |

*Table 12.* Comparison of congestion (RUDY estimator) for macro-only placements on the ISPD benchmark.

| Circuit | MaskPlace | WireMask-BBO | ChiPFormer | Diffusion (Ours) |
|---------|-----------|--------------|------------|------------------|
| adaptec1 | 312 | 139 | 140 | 149 |
| adaptec2 | 1068 | 1084 | 1180 | 668 |
| adaptec3 | 990 | 672 | 677 | 579 |
| adaptec4 | 945 | 793 | 779 | 584 |
| bigblue1 | 98.5 | 25.1 | 19.0 | 23.4 |
| bigblue2 | Timeout | 1924 | 500 | 523 |
| bigblue3 | 970 | 955 | 956 | 391 |
| bigblue4 | Timeout | 6290 | 2436 | 1451 |
| Average | — | 1485 | 836 | 546 |

*Table 13.* Comparison of runtime (minutes) on the IBM benchmark. ibm05 has been omitted because it contains no macros.

| Circuit | MaskPlace | WireMask-BBO | ChiPFormer | EfficientPlace | DREAMPlace | Diffusion (Ours) |
|---------|-----------|--------------|------------|----------------|------------|------------------|
| ibm01 | 154 | 209 | 98 | 54 | 0.308 | 1.85 |
| ibm02 | 165 | 204 | 87 | 61 | 0.411 | 2.25 |
| ibm03 | 123 | 217 | 75 | 61 | 0.393 | 2.17 |
| ibm04 | 63 | 208 | 82 | 61 | 0.401 | 2.21 |
| ibm06 | 34 | 224 | 80 | 29 | 0.229 | 2.38 |
| ibm07 | 58 | 223 | 80 | 63 | 0.261 | 2.87 |
| ibm08 | 75 | 207 | 105 | 79 | 0.260 | 3.38 |
| ibm09 | 50 | 221 | 71 | 46 | 0.257 | 3.08 |
| ibm10 | 516 | 228 | 236 | 268 | 0.455 | 4.50 |
| ibm11 | 79 | 224 | 106 | 80 | 0.303 | 3.40 |
| ibm12 | 390 | 253 | 196 | 206 | 0.469 | 5.24 |
| ibm13 | 93 | 225 | 127 | 95 | 0.613 | 4.19 |
| ibm14 | 393 | 266 | 187 | 216 | 0.760 | 5.95 |
| ibm15 | 83 | 254 | 113 | 86 | 0.923 | 6.81 |
| ibm16 | 107 | 217 | 137 | 130 | 0.784 | 7.88 |
| ibm17 | 489 | 266 | 250 | 358 | 0.839 | 10.33 |
| ibm18 | 60 | 216 | 93 | 58 | 0.742 | 8.06 |
| Average | 172 | 227 | 124 | 114 | 0.475 | 4.39 |

*Table 14.* Comparison of runtime (minutes) on the ISPD benchmark.

| Circuit | MaskPlace | WireMask-BBO | ChiPFormer | DREAMPlace | Diffusion (Ours) |
|---------|-----------|--------------|------------|------------|------------------|
| adaptec1 | 139 | 211 | 223 | 1.07 | 4.78 |
| adaptec2 | 195 | 209 | 234 | 1.34 | 4.53 |
| adaptec3 | 224 | 207 | 284 | 2.06 | 4.73 |
| adaptec4 | 718 | 212 | 467 | 2.43 | 4.94 |
| bigblue1 | 274 | 204 | 256 | 1.30 | 4.83 |
| bigblue2 | Timeout | 1396 | 5220 | 3.80 | 122 |
| bigblue3 | 648 | 233 | 494 | 3.14 | 5.13 |
| bigblue4 | Timeout | 596 | 1210 | 8.86 | 18.9 |
| Average | — | 408.5 | 1049 | 3.00 | 21.2 |

