# OpenReview forum: "Chip Placement with Diffusion Models"
_ICML.cc/2025/Conference — ICML 2025 poster_

### Official Review · Reviewer_izVP · 2025-02-24

**Overall Recommendation:** 2

**Summary:**

This paper proposes to address the challenges faced in RL-based placement methods, including 1) scalability to larger circuits, and 2) the trajectory cannot be reversed in RL-based methods. A method to synthesize placement data, and a diffusion-based method to tackle the placement task are proposed.

## update after rebuttal
After carefully reviewing the rebuttals and comments, I would like to maintain my current score.

**Claims And Evidence:**

Yes.

**Essential References Not Discussed:**

A fast and strong RL+MCTS method [1] for macro & mixed-size placement, which is not discussed and compared.

[1] Geng, Zijie, et al. "Reinforcement learning within tree search for fast macro placement." Forty-first International Conference on Machine Learning. 2024.

**Experimental Designs Or Analyses:**

Yes. There are no issues about the soundness/validity of any experimental designs or analyses.

**Methods And Evaluation Criteria:**

Yes.

**Other Comments Or Suggestions:**

1. Compare the inference speed of diffusion-based method and analytical method.
2. Consider other objectives, such as post-routing wire length, timing metrics, power, are hard to represent into a differentiable form, it is more attractive to inject these objectives into the optimization of diffusion-based methods, rather than merely concentrating on the optimization of HPWL.
3. For other suggestions, please refer to the weakness part.

**Other Strengths And Weaknesses:**

**Strengths**
A new approach to construct the synthesized data to alleviate the scarcity of data in the EDA field.

**Weaknesses**
1. Optimization of other objectives are not consideration, such as timing metric (WNS, TNS), final wirelength.
2. Constraints such as the non-overlap constraint cannot be guaranteed by the diffusion method, which need a post-processing legalization to further eliminate the overlap. in the contrary, overlap can be avoided in RL-based method like MaskPlace and ChipFormer through the filtering out invalid positions in the actions which lead to overlap, e.g., through the position mask proposed in MaskPlace.
3. The motivation of using diffusion-based method is not promising enough. The authors only discuss the weaknesses of RL-based methods, while the discussion of analytical methods such as DREAMPlace is ignored. The advantage of diffusion and the motivation should be further analyzed.
4. Performance between the proposed approach and DREAMPlace is very similar, with only decreasing from 23.6 to 22.7, while other cost, including the congestion, timing metrics (WNS & TNS), runtime (speed), and the resources cost for training the diffusion model are not compared.

**Questions For Authors:**

In Table 5, the magnitude of the mixed-size placement HPWL is 1e6, while in ChipFormer, the magnitude of mixed-size placement HPWL is 1e7 (shown in Table 8 in [1]). Could the authors to explain the inconsistency or check the correctness of the presented data?

[1] Lai, Yao, et al. "Chipformer: Transferable chip placement via offline decision transformer." International Conference on Machine Learning. PMLR, 2023.

**Relation To Broader Scientific Literature:**

The idea of constructing synthesized data to support the training of model is valuable.

**Theoretical Claims:**

Yes. There are no issues about the correctness of any proofs for theoretical claims.

---

> ### Author Rebuttal · Authors · 2025-04-01
>
> We thank the reviewer for the insightful feedback. Our response is as follows:
>
> **EfficientPlace:** Although EfficientPlace uses tree search to address the shortcomings of RL, it still requires significant training on every new circuit to perform well. We present additional experiments on the IBM benchmark below, showing that our method significantly outperforms EfficientPlace in both HPWL and congestion, while requiring a fraction of the time.
>
> **Table 1** Congestion and HPWL on IBM
> ||Wiremask-BBO|Chipformer|MaskPlace|EfficientPlace|Ours|
> |-|-|-|-|-|-|
> |Average Congestion (RUDY)|323.6|335.9|345.02|366.7|195.5|
> |Average HPWL (10^6) | 7.432|7.931|8.723|8.316|2.495|
>
> **DreamPlace:** We highlight that our method outperforms DreamPlace, a very strong baseline, while other macro placement methods fall far short. This could be because DreamPlace relies on gradient descent, which performs a local search, whereas the diffusion model can perform a global search based on the training data. We agree that this additional performance comes at a cost, with our method having a longer runtime as shown in Table 2 below, and requiring several days of training.
>
> Nevertheless, we emphasize that our work explores and develops a novel approach - training diffusion models - for the placement problem, and to our knowledge is the first to apply diffusion to this domain. Developing a method that is competitive with, even marginally outperforming, DreamPlace is still a significant improvement over prior macro placement approaches, and demonstrates that this approach has strong merits. We believe showing this is one of the contributions of our work.
>
> **Table 2** Runtimes in minutes
> ||Wiremask-BBO|ChipFormer|Dreamplace|Ours|
> |-|-|-|-|-|
> |IBM Average|227.2|124.88|0.475|4.39|
> |ISPD Average|886.5|1048.5|3.000|20.89|
>
> **Non-overlap:** While it is true that our method does not enforce hard constraints to prevent overlap, we find that in practice this is not an issue. Legality guidance, combined with gradient-based legalization, is effective in ensuring almost no overlaps in our macro placements.
>
> **Optimization of other objectives:** We presented results on congestion in table 1 above, which shows that our method achieves significantly lower congestion than the baselines. This result is consistent with findings from prior work [1] that finds congestion to be correlated with HPWL.
>
> We agree that downstream metrics such as PPA are important targets for optimization. However, optimizing for PPA is difficult, with many similar works focusing on simple proxy objectives like HPWL. Moreover, the commonly used benchmarks such as ISPD2005 and IBM do not support timing analysis. Because the goal of our work is to explore and develop a novel approach - training a diffusion model - to macro placement, we have therefore chosen to focus our contributions on developing the techniques necessary for such an approach, such as synthetic data generation, rather than simultaneously tackling PPA optimization. Therefore, while PPA optimization is an important end-goal, we leave it as an area for future work.
>
> **Answers to Questions:**
> 1. We believe the magnitude in the ChipFormer paper should be 1e5. Table 15 of the MaskPlace paper [2] has the same numbers as those in Table 8 of the ChipFormer paper, but with a scale of 1e5, which is the same as us.
>
> We hope we have been able to address your concerns.
>
>
>
> [1] Shi et. al. "Macro Placement by Wire-Mask-Guided Black-Box Optimization." Neurips, 2023.
>
> [2] Lai et. al. "MaskPlace: Fast Chip Placement via Reinforced Visual Representation Learning." Neurips, 2022.

---

### Official Review · Reviewer_m7a3 · 2025-03-12

**Overall Recommendation:** 3

**Summary:**

The authors propose a new diffusion-based method to address chip placement. Compared to existing RL approaches, pre-trained diffusion models can obtain the placement results on new circuits within minutes, which are much more efficient. After global placement, users can fix the positions of macros and optimize the cells using other cell placer like DREAMPlace.

## update after rebuttal
Based on the reply, I would like to increase the score to 3. I hope the authors could add these new experiments and illustrations to the revised manuscript for a more transparent presentation.

**Claims And Evidence:**

The authors claim that RL methods are slow and suffer from flexibility, however, they do not show diffusion models’ detailed time overhead.

**Essential References Not Discussed:**

Yes, I think the references are sufficient.

**Experimental Designs Or Analyses:**

The datasets used in the experiments are not in line with those in existing work. For example, this paper tests the approach on ibm01-18, but the baselines test the ispd05/25 benchmarks.

**Methods And Evaluation Criteria:**

Yes, the proposed method intuitively makes sense, but the procedure is not that clear in Sec. 4.3. I suggest the authors provide a detailed algorithm box.

**Other Comments Or Suggestions:**

- The authors can display placement visualizations of different methods on the same circuit.
- As the fixed-size performance significantly depends on cell placers (e.g, DREAMPlace), the authors could display more comparison results on macro-only settings.

**Other Strengths And Weaknesses:**

Other Strengths:
- The authors address chip placement under the perspective of diffusion models, which can obtain chip placement results on new circuits within minutes.
- The motivation is clear, as existing RL methods take a long time to complete the placement.

Other weaknesses:
- Non-learning methods, especially the DREAMPlace, are not included in the related work section.
- DREAMPlace has lots of versions with significantly different performance, so it is important for the authors to mention the version that they used in the experiments. I note that the authors mentioned DREAMPlace 4.1 but cited their paper in 2019.
- The datasets used in the experiments are not in line with those in existing work. The author could give a suitable explanation to address this weakness.

**Questions For Authors:**

- Why did the authors choose IBM benchmarks as the test dataset? I note that (all) the baselines the authors compared used ISPD05/15 as the datasets.
- The experimental results of HPWL on ibm01-04 are quite different from those displayed in ChipFormer. Is this because the results of ChipFormer are macro-only and the results in this work are mixed-size?
- What is the placement $x$? Is it the positions of all macros?

**Relation To Broader Scientific Literature:**

This paper is an approach for chip placement, which is in the field of physical design in electronic design automation.

**Theoretical Claims:**

No theoretical claim is provided.

---

> ### Author Rebuttal · Authors · 2025-04-01
>
> We thank the reviewer for their insightful feedback. Our response is as follows:
>
> **Choice of dataset:** We choose the IBM dataset for several reasons. First, it contains more circuits - 18, compared to 8 for ISPD2005. Second, the IBM dataset allows for easier comparison with other macro placement methods. The ISPD2005 benchmark contains circuits with a large number of macros (up to 23k), causing prior works to omit these circuits or pick macros to place according to various criteria. The IBM dataset avoids this issue, and allows for consistent evaluation of macro placement methods on all 18 circuits.
> Nevertheless, we have evaluated our method on the ISPD2005 dataset, using the macro-only setting for easier comparison, with the results shown in Table 1 below. Our method achieves significantly improved performance over the baselines.
>
> **Macro-only Evaluations:** We agree with this suggestion, and present our results in the tables below.
>
> **Table 1** HPWL on ISPD2005 benchmark
> ||Wiremask-BBO|Chipformer|MaskPlace|Ours|
> |-|-|-|-|-|
> |Average HPWL (10^6) |19.534|13.689|14.925|4.393|
>
> **Table 2** HPWL on IBM benchmark
> ||Wiremask-BBO|Chipformer|MaskPlace|EfficientPlace|Ours|
> |-|-|-|-|-|-|
> |Average HPWL (10^6) | 7.432|7.931|8.723|8.316|2.495|
>
> **Time overhead:** We present the runtimes for our method and baselines in the table below. Our method is significantly faster than other macro placement methods, taking on average 4 and 20 minutes on the IBM and ISPD benchmarks respectively, compared to RL or BBO methods that take 10 times longer. We note however that we are slower than Dreamplace, and further optimization of our code and diffusion sampling is an interesting area of future work.
>
> **Table 3** Runtimes in minutes
> ||Wiremask-BBO|ChipFormer|Dreamplace|Ours|
> |-|-|-|-|-|
> |IBM Average|227.2|124.88|0.475|4.39|
> |ISPD Average|886.5|1048.5|3.000|20.89|
>
> To clarify, we used Dreamplace 4.1, and will correct the citation to reflect this.
>
> **Answers to questions:**
>
> 1. Our motivation for choosing IBM is detailed above. We have also performed additional experiments on the ISPD05 benchmark, with results shown in Table 1 above.
>
> 2. Yes, Table 2 in the ChipFormer paper reports macro-only HPWL, whereas Table 5 in our paper reports mixed-size HPWL.
>
> 3. x is a (V x 2) array, where V is the number of objects in the netlist, containing the 2D positions of all objects, which includes macros and standard cell clusters.
>
> We hope we have been able to address your concerns.

---

> > ### Comment · Reviewer_m7a3 · 2025-04-02
> >
> > Thanks for the authors' response. Though some of my concerns have been addressed, I still have some questions regarding the performance of baselines.
> >
> > First, the HPWL performances of MaskPlace and ChiPFormer on mixed-size placement in this paper are different from those in their original papers. For example, according to MaskPlace, the HPWL on ibm01 is $24.18\times 10^5$. In ChiPFormer, the HPWL on ibm01 is $16.70\times 10^7$ (it might be a typo in their paper if I understand correctly, should be $16.70\times 10^5$). However, in your paper, these two values are $3.33\times 10^6$ and $3.35\times 10^6$. (same issues also occur in other benchmarks ibm02, ibm03...) What is the reason for this discrepancy?
> >
> > Second, for the newly-added macro-only experiments performed on the ISPD2005 benchmark, such circumstances may also exist. Additionally, I think it is not proper to only show the average HPWL or time for the ISPD2005 benchmark as the circuits differ significantly in their scale. The authors could detail the performance of **each circuit**.

---

> > > ### Author Response · Authors · 2025-04-04
> > >
> > > We thank the reviewer for your thoughtful response, and hope the following addresses your remaining concerns.
> > >
> > > **Mixed-size HPWL on IBM:** The differences in HPWL from the original papers can be accounted for by differences in evaluation setups. The most major is whether the macros are fixed during standard cell placement. For ease of comparison, and to make clearer the impact of the initial macro placements, we fixed the macro positions when placing standard cells with DreamPlace, while many prior works allowed them to move. Another difference is the DreamPlace version (we use 4.1).
> > >
> > > **Macro-only HPWL on ISPD:** The differences in HPWL from the original papers is because some baselines select and place only a subset of the macros (selection criteria differs between baselines), while we place all macros for all baselines to ensure a fair comparison. This is especially significant for bigblue2 and bigblue4, which have large numbers of macros, but can also apply to other circuits. MaskPlace, for instance, places only 128 macros [1] for adaptec1.
> > >
> > > **Reporting performance for each circuit:** We present the per-circuit figures in the tables below, with HPWL in Tables 1 & 2, runtimes in Tables 3 & 4, and congestion (requested by other reviewers and included for reference) in Tables 5 & 6. We note some minor differences (Tables 1, 3, 5) with the earlier reported averages on the ISPD benchmark. We erroneously copied the data previously, and deeply apologize for our mistake. The tables below contain the corrected figures. Nevertheless, we emphasize that our conclusions remain unaffected: our method produces significantly better results on both HPWL and congestion on the IBM and ISPD benchmarks, while running faster than prior methods (except DreamPlace).
> > >
> > > We hope we have been able to address your concerns.
> > >
> > > [1] See line 51 and 63 of PPO2.py in the MaskPlace public github repository.
> > >
> > > **Table 1** HPWL ($\times 10^5$) on ISPD.
> > > ||MaskPlace|Wiremask-BBO|Chipformer|Ours|
> > > |-|-|-|-|-|
> > > |adaptec1|8.57|5.81|6.75|9.19|
> > > |adaptec2|77.7|54.5|63.8|31.0|
> > > |adaptec3|108|59.2|73.2|54.4|
> > > |adaptec4|91.9|62.7|85.8|54.5|
> > > |bigblue1|3.11|2.12|3.05|2.64|
> > > |bigblue2|Timeout|186|85.8|38.8
> > > |bigblue3|84.0|66.2|79.2|35.9
> > > |bigblue4|Timeout|798|548|141
> > > |Average|-|154|116|45.9
> > >
> > > **Table 2** HPWL ($\times 10^5$) on IBM.
> > > ||MaskPlace|Wiremask-BBO|Chipformer|EfficientPlace|Ours|
> > > |-|-|-|-|-|-|
> > > |ibm01|4.30|2.78|3.88|3.66|1.16
> > > |ibm02|5.54|4.19|5.05|4.42|2.68
> > > |ibm03|3.31|3.30|3.74|3.87|1.07
> > > |ibm04|6.91|5.43|5.96|6.10|2.40
> > > |ibm06|0.93|0.85|0.87|0.84|0.32
> > > |ibm07|2.67|2.66|2.36|3.42|0.78
> > > |ibm08|20.6|19.2|19.9|19.3|9.32
> > > |ibm09|2.45|1.76|1.77|2.57|0.44
> > > |ibm10|23.8|18.2|18.2|20.6|5.28
> > > |ibm11|4.15|3.75|3.25|4.70|0.78
> > > |ibm12|14.9|11.8|13.0|12.1|2.85
> > > |ibm13|4.58|4.41|4.02|5.37|1.05
> > > |ibm14|8.43|9.80|7.44|11.7|2.42
> > > |ibm15|4.68|7.77|2.67|5.98|1.06
> > > |ibm16|18.3|14.8|15.5|15.2|6.11
> > > |ibm17|16.8|12.2|13.7|17.9|3.20
> > > |ibm18|5.98|3.44|4.19|3.64|1.52
> > > |Average|8.72|7.43|7.33|8.32|2.49
> > >
> > > **Table 3** Runtime (minutes) on ISPD.
> > > ||MaskPlace|Wiremask-BBO|Chipformer|Dreamplace|Ours|
> > > |-|-|-|-|-|-|
> > > |adaptec1|139|211|223|1.07|4.78
> > > |adaptec2|195|209|234|1.34|4.53
> > > |adaptec3|224|207|284|2.06|4.73
> > > |adaptec4|718.2|212|467|2.43|4.94
> > > |bigblue1|274|204|256|1.30|4.83
> > > |bigblue2|-|1396|5220|3.80|122
> > > |bigblue3|648|233|494|3.14|5.13
> > > |bigblue4|-|596|1210|8.86|18.9
> > > |Average|-|408.5|1049|3.00|21.2
> > >
> > > **Table 4** Runtime (minutes) on IBM.
> > > ||MaskPlace|Wiremask-BBO|Chipformer|EfficientPlace|Dreamplace|Ours|
> > > |-|-|-|-|-|-|-|
> > > |ibm01|154|209|98|54|0.308|1.85
> > > |ibm02|165|204|87|61|0.411|2.25
> > > |ibm03|123|217|75|61|0.393|2.17
> > > |ibm04|63|208|82|61|0.401|2.21
> > > |ibm06|34|224|80|29|0.229|2.38
> > > |ibm07|58|223|80|63|0.261|2.87
> > > |ibm08|75|207|105|79|0.260|3.38
> > > |ibm09|50|221|71|46|0.257|3.08
> > > |ibm10|516|228|236|268|0.455|4.50
> > > |ibm11|79|224|106|80|0.303|3.40
> > > |ibm12|390|253|196|206|0.469|5.24
> > > |ibm13|93|225|127|95|0.613|4.19
> > > |ibm14|393|266|187|216|0.760|5.95
> > > |ibm15|83|254|113|86|0.923|6.81
> > > |ibm16|107|217|137|130|0.784|7.88
> > > |ibm17|489|266|250|358|0.839|10.33
> > > |ibm18|60|216|93|58|0.742|8.06
> > > |Average|172|227|124|114|0.475|4.39
> > >
> > > **Table 5** Congestion on ISPD
> > > ||MaskPlace|Wiremask-BBO|Chipformer|Ours|
> > > |-|-|-|-|-|
> > > |adaptec1|312|139|140|149
> > > |adaptec2|1068|1084|1180|668
> > > |adaptec3|990|672|677|579
> > > |adaptec4|945|793|779|584
> > > |bigblue1|98.5|25.1|19.0|23.4
> > > |bigblue2|-|1924|500|523
> > > |bigblue3|969.8|955|956|391
> > > |bigblue4|-|6290|2436|1451
> > > |Average|-|1485|836|546
> > >
> > > **Table 6** Congestion on IBM
> > > ||MaskPlace|Wiremask-BBO|Chipformer|EfficientPlace|Ours|
> > > |-|-|-|-|-|-|
> > > |ibm01|289|253|266|316|160
> > > |ibm02|228|243|205|257|178
> > > |ibm03|176|173|173|214|117
> > > |ibm04|449|483|490|480|260
> > > |ibm06|79.2|77.1|76.9|76.7|42.8
> > > |ibm07|154|164|160|177|83.0
> > > |ibm08|1232|1198|1261|1288|776
> > > |ibm09|127|119|111|153|49.1
> > > |ibm10|480|463|466|538|362
> > > |ibm11|180|183|172|240|69.2
> > > |ibm12|392|212|357|360|190
> > > |ibm13|163|202|177|209|86.0
> > > |ibm14|378|375|378|418|232
> > > |ibm15|162|173|173|227|69.2
> > > |ibm16|574|497|528|534|334
> > > |ibm17|531|464|488|483|204
> > > |ibm18|271|221|229|266|111
> > > |Average|345|324|336|367|196

---

### Official Review · Reviewer_QY6P · 2025-03-14

**Overall Recommendation:** 3

**Summary:**

The authors proposed a diffusion model-based chip placement strategy. They also developed a novel data generation algorithm and a synthetic dataset, training the model to enable zero-shot transfer to real circuits. Additionally, they introduced a neural network model that demonstrates strong performance and scalability.

**Claims And Evidence:**

The major claims by the authors:
1. Synthetic data generation: The approach generates a plausible netlist ensuring that the given placement is near-optimal while enabling data generation without relying on commercial tools or higher-level design specifications. Tables 1, 2, and 3 provide evidential support for this claim.

2. Dataset design: An extensive empirical study was conducted to investigate the generalization properties of models trained on synthetic data, identifying several factors—such as the scale parameter—that contribute to poor generalization. These insights were utilized to design synthetic datasets that enable effective zero-shot transfer to real circuits. Once again, Tables 1, 2, 3, and 7 provide evidential support for this claim.

3. Model architecture: The authors proposed a novel neural network architecture incorporating interleaved graph convolutions and attention layers, resulting in a model that is both computationally efficient and highly expressive. Tables 5 and 6 provide support for this claim.

**Essential References Not Discussed:**

NA

**Ethical Review Concerns:**

No ethical review concerns noticed.

**Experimental Designs Or Analyses:**

Yes....... soundness/validity of experimental designs has been validated against:
1. Synthetic data generation: The approach generates a plausible netlist ensuring that the given placement is near-optimal while enabling data generation without relying on commercial tools or higher-level design specifications. Tables 1, 2, and 3 provide evidential support for this claim.

2. Dataset design: An extensive empirical study was conducted to investigate the generalization properties of models trained on synthetic data, identifying several factors—such as the scale parameter—that contribute to poor generalization. These insights were utilized to design synthetic datasets that enable effective zero-shot transfer to real circuits. Once again, Tables 1, 2, 3, and 7 provide evidential support for this claim.

3. Model architecture: The authors proposed a novel neural network architecture incorporating interleaved graph convolutions and attention layers, resulting in a model that is both computationally efficient and highly expressive. Tables 5 and 6 provide support for this claim.

**Methods And Evaluation Criteria:**

The proposed method is thoroughly evaluated using a well-designed experimental setup and relevant metrics. The authors provide an in-depth analysis, effectively demonstrating proof-of-concept to support their claims. Additionally, the combination of proposed strategies enables the generation of placements for unseen netlists in a zero-shot manner, achieving competitive performance with state-of-the-art (SOTA) methods on the IBM benchmark dataset ICCAD04.

**Other Comments Or Suggestions:**

The authors should benchmark the proposed approach on other private or public datasets to establish its generalization and scalability, such as any modern IC design netlists.

**Other Strengths And Weaknesses:**

Mentioned and discussed in previous sections as "Methods And Evaluation Criteria*" and "Experimental Designs Or Analyses*".

**Questions For Authors:**

I would like to hear the authors' thoughts on conducting an additional experiment using other private or public datasets to establish the generalization and scalability of the approach, such as modern IC design netlists.

**Relation To Broader Scientific Literature:**

The research topic and the presented idea, despite certain limitations, are interesting and hold potential significance for the broader research community. This is particularly true from two perspectives: synthetic data generation and dataset design. Additionally, the combination of the proposed strategies enables the generation of placements for unseen netlists in a zero-shot manner, achieving competitive performance with state-of-the-art (SOTA) methods on the IBM benchmark dataset ICCAD04.

**Theoretical Claims:**

Yes, the theoretical claims regarding the quality of synthetic data generation, dataset design, scalability, and generalization impact have been quantified through experimental validation.

---

> ### Author Rebuttal · Authors · 2025-04-01
>
> We thank the reviewer for their helpful feedback. Our response is as follows:
>
> **Additional benchmarks:** We have included experiments on the ISPD2005 benchmark, which we show in the table below. To facilitate comparison with baselines, we follow the suggestion of reviewer m7a3 and present HPWL and congestion in the macro-only setting. These results show that our method significantly outperforms baselines on this benchmark as well.
>
> **Table 1** Congestion and HPWL of macro placements on ISPD2005
> ||Wiremask-BBO|Chipformer|MaskPlace|Ours|
> |-|-|-|-|-|
> |Average Congestion (RUDY)|1837|988.7|1291|539.4|
> |Average HPWL (10^6) |19.534|13.689|14.925|4.393|
>
> We hope we have been able to address your concerns.

---

### Official Review · Reviewer_KAy5 · 2025-03-20

**Overall Recommendation:** 3

**Summary:**

This paper applies diffusion models to macro placement. The motivation is that existing RL-based methods for macro placement are slow and lack flexibility. To provide more data for training, this paper generates synthetic data by randomly placing objects, sampling pins, and creating edges based on a distance-dependent probability. The model architecture combines GNN and attention layers, with MLP blocks and sinusoidal encodings. Guided sampling is used to optimize placement quality. Experiments on synthetic data and the ICCAD04 benchmark show that the model can achieve competitive results in terms of legality and HPWL, and it performs well in mixed-size placement.

**Claims And Evidence:**

“WireMask-BBO must be started from scratch for each new circuit.” Incorrect claim. The original WireMask-BBO paper show that it can fine-tune existing placements. I think the one of drawbacks of WireMask-BBO is the generalization ability and search efficiency, compared to those learning-based RL approaches.

**Essential References Not Discussed:**

There are many recent papers on reinforcement learning for chip placement in AI conferences [1-3], which I believe should be at least discussed.

[1] Reinforcement Learning within Tree Search for Fast Macro Placement. ICML'24.

[2] Reinforcement Learning Policy as Macro Regulator Rather than Macro Placer. NeurIPS'24.

[3] LaMPlace: Learning to Optimize Cross-Stage Metrics in Macro Placement. ICLR'25.

**Experimental Designs Or Analyses:**

1. I believe the validity of synthetic data requires more discussion. If synthetic data is a contribution, some experiments should be included to demonstrate that the previous method (e.g., Flora) is ineffective.
2. Followed by point 1, I believe an important contribution is the study of the role of synthetic placement datasets. If the dataset proposed in this paper truly "covers the important features" as stated in line 311, it should also improve the reinforcement learning methods; however, the authors have not compared this aspect.
3. Clustering standard cells: How does it compare to placing only several macros and then using DMP to place standard cells? How does it compare to other RL methods with the same clustering approaches?

**Methods And Evaluation Criteria:**

There is no PPA evaluation. Although wire length is important, many articles have actually found that its impact on the final result is also limited. Recently, there are some open-source platforms have provided PPA evaluation such as OpenRoad and [1].

I believe adding PPA assessment could significantly enhance the quality of this paper.

[1] Benchmarking End-To-End Performance of AI-Based Chip Placement Algorithms. arxiv, 2024.

**Other Comments Or Suggestions:**

1. More references should be added; for example, the first paragraph of the introduction has no references at all. It is necessary to include more articles on EDA background so that people in the machine learning field can understand the context of the problem.
2. The third contribution - Model Architecture, cannot be considered a contribution, as it does not seem novel to me since many papers apply similar architectures. It would be better to list the application of diffusion for chip placement as a contribution here.
3. Please use "DMP" rather than "DP" in Table 5 to represent DREAMPlace.

**Other Strengths And Weaknesses:**

Strengths:
1. Completely using synthetic data has demonstrated good generalization ability.
2. The authors studied the impact of synthetic data on generalization and conducted extensive analyses, including the number of edges and vertices, etc.

Weaknessess:
1. The writing should be improved. Besides, adding some discussions of the background and recent related works would also be very beneficial.

**Questions For Authors:**

1. Where were the other RL methods in Table 5 trained?
2. How are overlaps handled? Is the legalization method provided in DREAMPlace used?
3. Fig. 6: Increasing the scale parameter significantly causes legality to violate constraints. Would using a diverse range of scales (e.g., randomly samling in a large range) during training lead to better results?

**Relation To Broader Scientific Literature:**

Chip placement is an vital tasks to EDA. Previous chip placement relies on RL and suffers several limitations. This paper first propose to use diffusion model to conduct chip placement and perform well.

**Theoretical Claims:**

No theory part.

---

> ### Author Rebuttal · Authors · 2025-04-01
>
> Thank you for your insightful feedback and suggestions. We hope the following can address your concerns.
>
> **PPA evaluation:** We agree that PPA evaluation and optimization is important. As a step towards analyzing and optimizing downstream objectives, we also evaluated congestion of our macro placements, with the results in Table 1 below showing that our method significantly outperforms the baselines not just in HPWL, but in congestion.
>
> However, optimizing for PPA is difficult, with many similar works focusing on simple proxy objectives like HPWL. Moreover, the commonly used benchmarks such as ISPD2005 and IBM do not support timing analysis. Because the goal of our work is to explore and develop a novel approach - training a diffusion model - to macro placement, we have therefore chosen to focus our contributions on developing the techniques necessary for such an approach, such as synthetic data generation, rather than simultaneously tackling PPA optimization. We believe that despite its shortcomings, HPWL is a reasonable optimization objective, particularly as a first step when exploring a new approach.
>
> Therefore, while PPA optimization is an important end-goal, we leave it as a direction for future work.
>
> **Table 1** Congestion on IBM macro placements.
> ||Wiremask-BBO|Chipformer|MaskPlace|EfficientPlace|Ours|
> |-|-|-|-|-|-|
> |Average Congestion (RUDY)|323.6|335.9|345.02|366.7|195.5|
>
> **Validity of synthetic data:** We trained different-sized models on a dataset generated using Flora’s algorithm and found that models trained on their dataset show much poorer legality than ours when evaluated on the clustered IBM circuits. This indicates that the Flora dataset generalizes poorly, in contrast to ours. The results below are after 1M training steps.
>
> **Table 2** Performance of Large and Medium models trained on different datasets.
> ||Large+Flora|Medium+Flora|Large+v1 (Ours)|Medium+v1 (Ours)|
> |-|-|-|-|-|
> |Legality|0.283|0.349|0.806|0.784|
> |HPWL (10^7)|3.058|3.306|3.330|3.527|
>
> **Training RL with synthetic data:** This is a good point, and we believe this to be an interesting experiment to perform in the future.
>
> **Clustering standard cells:** We performed mixed-size placement on the IBM benchmark using clustered standard cells (our approach), and the suggested approach (also commonly used in literature) of first placing macros only. The results below show that using clustered standard cells performs better, likely because the macro positions can be informed by connectivity and space needed for standard cells.
>
> **Table 3** Mixed-size placement performance with and without standard cell clusters.
> ||Clustering|Placing Macro-only|
> |-|-|-|
> |Average HPWL (10^6)|22.7|27.9|
>
> **Recent papers:** We have conducted additional experiments comparing with EfficientPlace[1] in the macro-only setting, which we show below. Although EfficientPlace uses tree search to address the shortcomings of RL, our method still produces higher-quality samples in HPWL, while requiring a fraction of the sampling time.
>
> **Table 4** Comparison of various methods, including EfficientPlace, on the IBM benchmark.
> ||Wiremask-BBO|Chipformer|MaskPlace|EfficientPlace|Ours|
> |-|-|-|-|-|-|
> |Average HPWL (10^6) | 7.432|7.931|8.723|8.316|2.495|
>
> **Answers to questions:**
> 1. We used the official implementations and trained on the test (ie. IBM benchmark) circuits.
> 2. We post-processed the macro placements with our own gradient-based legalizer. Combined with legality guidance, this method is effective in ensuring almost no overlaps.
> 3. As mentioned in section 5.1.4, we do use a diverse range of scales to generate our dataset, sampling the scale from a log-uniform distribution, with a range of (0.05, 1.6) and (0.025, 0.8) for the v1 and v2 datasets respectively.
>
> We thank the reviewer for the helpful comments on the writing and clarity of our paper, and will be sure to make the necessary changes. We hope we have been able to address your concerns.
>
> [1] Reinforcement Learning within Tree Search for Fast Macro Placement. ICML'24.

---

### Decision · Program_Chairs · 2025-05-01

**Decision:**

Accept (poster)

**Comment:**

**Summary**: Paper used the standard DDPM diffusion pipeline  with a denoiser whose architecture has interleaved attention and Graph neural Net layers and many other modifications to adapt to the problem of placing macros (large components as I understand it ) on 2d Chip layouts. A lot of prior works use RL algorithms where they sequentially commit to placement of components with RL rewards reinforcing the policy. Authors point out that this problem calls for a simultaneous placement and therefore diffusion could play a role.

Authors come up with a simple method that randomly generates rectangular components and places them randomly obeying some overlap constraints and placement of pins satisfy some wire-length constraints. Authors show that by training a DDPM model with a novel architecture for denoiser using GNN layers, it shows impressive performance on many downstream metrics on real world benchmarks.

**Discussion/Rebuttal points**
1) From the discussion, it is acknowledged by reviewers and authors that this method is very competitive with DREAMplace - a state of the art method that uses a lot of domain specific information to optimize placements. Given that it is trained only on synthetic data and guided at test time by differentiable proxies, I see value in this being competitive with a good custom prior alternative.

2) Authors acknowledge that the benchmarks they have chosen are not compatible with other metrics like timing analysis but congestion based metrics are used to evaluate in the rebuttal. Different synthetic datasets were tried and their dataset generation method seems to outperform.

**Overall**: Main weakness is lack of algorithmic novelty - its the same standard DDPM pipeline. However, the architecture for denoiser is novel and tailored to this domain and that is the main punchline. I don't consider lack of all downstream metrics in evaluation to be a potential block against publication. For the potential impact of diffusion to an important domain, novel architectural choices and demonstration that purely pre-training on synthetic data suffices, I am recommending accept.

Note to authors: Please include all the experimental results from rebuttal (including on new metrics and ablations on other synthetic dataset) to the paper.